# The Metformin Mechanism on Gluconeogenesis and AMPK Activation: The Metabolite Perspective

**DOI:** 10.3390/ijms21093240

**Published:** 2020-05-03

**Authors:** Loranne Agius, Brian E. Ford, Shruti S. Chachra

**Affiliations:** Biosciences Institute, The Medical School, Newcastle University, Newcastle upon Tyne NE2 4HH, UK; brian.ford2@ncl.ac.uk (B.E.F.); Shruti.chachra@ncl.ac.uk (S.S.C.)

**Keywords:** Liver metabolism, gluconeogenesis, AMPK, metformin, phosphofructokinase-1

## Abstract

Metformin therapy lowers blood glucose in type 2 diabetes by targeting various pathways including hepatic gluconeogenesis. Despite widespread clinical use of metformin the molecular mechanisms by which it inhibits gluconeogenesis either acutely through allosteric and covalent mechanisms or chronically through changes in gene expression remain debated. Proposed mechanisms include: inhibition of Complex 1; activation of AMPK; and mechanisms independent of both Complex 1 inhibition and AMPK. The activation of AMPK by metformin could be consequent to Complex 1 inhibition and raised AMP through the canonical adenine nucleotide pathway or alternatively by activation of the lysosomal AMPK pool by other mechanisms involving the aldolase substrate fructose 1,6-bisphosphate or perturbations in the lysosomal membrane. Here we review current interpretations of the effects of metformin on hepatic intermediates of the gluconeogenic and glycolytic pathway and the candidate mechanistic links to regulation of gluconeogenesis. In conditions of either glucose excess or gluconeogenic substrate excess, metformin lowers hexose monophosphates by mechanisms that are independent of AMPK-activation and most likely mediated by allosteric activation of phosphofructokinase-1 and/or inhibition of fructose bisphosphatase-1. The metabolite changes caused by metformin may also have a prominent role in counteracting G6pc gene regulation in conditions of compromised intracellular homeostasis.

## 1. Introduction: The Metformin Mechanism and AMPK Activation

Type 2 diabetes is a chronic disease of increasing prevalence in association with obesity and positive energy balance. It manifests as a progressive rise in blood glucose and increased hepatic glucose production by gluconeogenesis. Metformin (dimethyl biguanide) is the most commonly prescribed therapy for this condition and one of its main therapeutic effects is the inhibition of hepatic gluconeogenesis [1,2]. This may be elicited through either acute inhibition of gluconeogenic flux or through chronic changes in gene expression. Despite several decades of research on the mechanisms by which metformin and other biguanides that preceded it in diabetes therapy, elicit inhibition of gluconeogenesis there remains much debate on which of the candidate mechanisms has a prominent role in its therapeutic effects.

One of the most extensively studied mechanisms for metformin is the activation of the signalling kinase AMPK [3,4]. The canonical mechanism involves accumulation of metformin in mitochondria by virtue of its positive charge at physiological pH causing inhibition of the respiratory chain at the level of Complex 1 [5,6,7] resulting in compromised adenine nucleotide phosphorylation potential and a rise in AMP promoting activation of the energy sensor, AMPK. This in turn phosphorylates a range of target proteins [8,9]. AMPK is a serine threonine kinase that was first identified as a negative regulator of hepatic acetyl-CoA carboxylase and HMG-CoA reductase, key regulators of fatty acid and cholesterol biosynthesis [10]. Its role in the inhibition of de novo fatty acid synthesis is firmly established and accordingly there is good consensus for AMPK activation accounting for the inhibition of fatty acid synthesis by metformin [3,4]. But a key outstanding issue is whether activation of AMPK has a role in the blood glucose lowering effect of metformin and if so to what extent relative to other mechanisms that manifest at low therapeutic doses of metformin [4,11]. A second question is whether the activation of AMPK by metformin is mediated by inhibition of Complex 1 or by other mechanisms that may manifest at lower metformin concentrations, which have negligible effect on cell adenine nucleotide levels [4]. Although AMPK is conventionally regarded as an energy sensor, recent studies identified fructose 1,6-bisphosphate (F1,6P_2_) as a metabolite regulator of AMPK [12,13,14]. This intermediate of glycolysis and gluconeogenesis links the PFK1/FBP1 cycle and aldolase and is present in liver at substantially lower concentrations than proximal and distal intermediates. Because of its low concentration there is little data on the effects of low metformin doses on this metabolite. The purpose of this paper is to discuss current understanding of the metformin mechanism on gluconeogenesis and of AMPK activation in relation to linked changes in cellular metabolites that may control enzyme activity through either allosteric mechanisms or control of gene expression.

## 2. AMPK Activation

### 2.1. Control by Adenine Nucleotides, Calcium and F1,6P_2_

AMPK is a heterotrimer comprising a catalytic subunit (α) and two regulatory (β and γ) subunits. These subunits are encoded by seven genes giving rise to α1, α2 (*Prkaa1,2*), β1,β2 (*Prkab1,2*), γ1,γ2γ3 (*Prkag1,2,3*) which enable 12 combinations of units and thereby potential diversity of response to cater for tissue-specific functions. AMPK is activated allosterically by binding of AMP to the γ subunit and by phosphorylation of Thr^172^ within an activation loop in the α-subunit [8]. The upstream kinases include LKB1 (Liver kinase B1) and CaMKK2 (Ca^2+^-calmodulin dependent protein kinase 2) [15,16,17,18]. LKB1 is constitutively active but is recruited to AMPK by allosteric mechanisms linked to adenine nucleotide binding to the γ subunit or by F1,6P_2_ dissociation from aldolase which promotes formation of a complex linking LKB1 and AMPK at the lysosomal surface. CaMKK2 is activated by an increase in calcium. Effects of metformin on agonist-induced Ca^2+^ oscillations and Ca^2+^ release from intracellular stores have been reported but not in connection with AMPK activity [19]. A third kinase for Thr^172^ phosphorylation is TAK1 (transforming growth factor β activated kinase-1) which remains as yet a contentious candidate after LKB1 and CAMKK2 [20]. However a role for TAK1 in activation of lysosomal AMPK has been reported [21,22] and a liver-selective TAK1 deletion mouse model is characterized by dysregulation of both lipid metabolism and autophagy consistent with compromised AMPK regulation [23].

The β subunits comprise myristoylated N-termini (which target AMPK to lysosomes and mitochondria), carbohydrate-binding domains (β-CBD), β-linkers and C-terminal domains (β-CTD) which constitute part of the binding sites of allosteric activators (A-769667, 991). The γ subunits are stabilized in the AMPK complex through an interdomain β-sheet formed with the β-CTD. They comprise 4 repeats of a motif termed the cystathionine β-synthase domain. One such domain binds AMP constitutively and another two bind AMP, ADP or ATP with different affinity enabling allosteric activation by the AMP/ATP ratio [8,9]. Binding of AMP and ADP promote phosphorylation of Thr^172^ by LKB1 and more modestly by CAMKK2 [24,25] and both AMP and ADP restrict dephosphorylation of Thr^172^ by protein phosphatase 2C [25,26]. However only AMP promotes allosteric activation [26]. The fractional activation by AMP is greater for γ1 and γ2 than for γ3 at physiological ATP [25]. Isoform diversity is also apparent from subcellular location, since α2 but not α1 complexes localize to the nucleus [27].

Although AMPK in all subcellular pools (cytoplasmic, lysosomal, mitochondrial and nuclear) can be regulated by adenine nucleotides through the canonical pathway, other “non-canonical” mechanisms for the lysosomal pool are triggered by either changes in F1,6P_2_ which affects LKB1 mediated phosphorylation [12] or Galectin-9 recruitment which promotes TAK1 mediated phosphorylation [21,22]. A key function of lysosomal AMPK is the stimulation of autophagy which is the process of degradation of long-lived proteins and macromolecular structures for recycling. The lysosomes are the ultimate destination for this process and harbour complex signalling pathways for the reciprocal control of AMPK and mTORC1, a major driver of protein homeostasis [28]. Activation of lysosomal AMPK by low glucose is triggered by depletion of F1,6P_2_ the substrate of FBP1 and aldolase, which promotes formation of a multiprotein complex at the lysosome comprising aldolase, the vacuolar H^+^-ATPase, the pentameric Ragulator Complex and the scaffolding protein AXIN which binds both AMPK and LKB1 [12,14]. Activation of cytoplasmic AMPK is contingent on moderately raised AMP (2–3 fold) and formation of a complex comprising AXIN, LKB1 and AMPK [14]. Activation of mitochondrial AMPK is dependent on greater elevation of AMP (~6-fold), which promotes phosphorylation of AMPK by LKB1 independently of AXIN [14]. The functional consequence of this hierarchy of AMP thresholds is that moderately compromised phosphorylation potential which activates lysosomal and cytoplasmic but not mitochondrial AMPK promotes inhibition of lipogenesis and protein synthesis by targeting ACC1, SREBP-1c and mTORC1 whereas more substantial ATP depletion is required to enhance mitochondrial fatty acid oxidation (by phosphorylating ACC2) or mitochondrial fragmentation through phosphorylation of MFF (mitochondrial fission factor) which is the receptor for dynamin like protein-1 on the mitochondrial outer membrane [29,30,31].

### 2.2. Changes Linked to Nutritional State or the Gluconeogenic/Glycolytic Poise

Liver AMPK is activated by fasting and inactivated by refeeding with the α1 isoform showing a greater and more rapid response than α2 [32]. Insulin and glucagon do not seem to be involved [33], however changes in the ATP/ADP ratio and thereby AMP as well as F1,6P_2_ (see below) are all candidate regulators. Low glucose is often implicated as the trigger for AMPK activation [9], however in liver the mechanistic link may be the change in directionality of flux from glycolysis in the absorptive state to gluconeogenesis during fasting. Ethanol is a strong negative regulator of liver AMPK causing both a decrease in Thr^172^ phosphorylation and a decline in the α-unit protein level. Proposed mechanisms include oxidative stress and ceramides [34,35]. Whether the more reduced cytoplasmic NAD/NADH redox state is involved in the negative regulation by ethanol and refeeding through consequent metabolite changes linked to the dehydrogenase equilibria remains to be tested. In hepatocytes showing higher AMPK activity in a glucose-free medium with gluconeogenic precursors compared with high glucose [33], candidate mechanisms may involve changes in compartmentation of enzymes of glycolysis and gluconeogenesis. Possible candidates include: glucokinase which translocates from the nucleus to the cytoplasm [36] and aldolase and FBP1 which colocalise at low glucose but not at high glucose when FBP1 translocates to the nucleus [37]. The interaction between FBP1 and aldolase is unique to the liver isoform (aldolase-B) and is regulated by AMP, F6P, F2,6P_2_, F1,6P_2_. A proposed hypothesis is that there are distinct pools of aldolase-B linked to glycolysis or gluconeogenesis [37]. This implies different occupancy by F1,6P_2_ and thereby differential involvement of aldolase pools in AMPK activation.

### 2.3. Activation by Biguanides and Mitochondrial Inhibitors (The Canonical Pathway)

Biguanides and rotenone (a Complex 1 inhibitor), cause activation of AMPK that parallels the compromised phosphorylation potential [38,39,40], taking into account that changes at low metformin may be at the limits of assay detection. Ota and colleagues [40] expressed the yeast internal NADH-quinone oxidoreductase (NDI1) and provided supportive evidence that Complex 1 inhibition accounts for the metformin activation of AMPK, a conclusion endorsed by other approaches [41]. NDI1 is one of two yeast NADH-Q oxidoreductases and is both rotenone-insensitive and lacks a proton translocase. NDI1 faces the mitochondrial matrix unlike the external isoform that faces the intermembrane space. The latter serves to oxidize cytoplasmic NADH because yeast lacks shuttles for reducing equivalents. NDI1 can substitute for functional defects in mammalian Complex 1 because it localizes in the correct orientation in mitochondria, when expressed in mammalian cells [42]. NDI1 is thereby, a very useful tool to test for Complex 1 involvement as distinct from other candidate mitochondrial targets such as ATP-synthase [7] in the metformin mechanism.

### 2.4. Metformin Activation of Lysosomal AMPK (Non-Canonical Pathways)

Zhang and colleagues [43] showed, using mouse knock-down models that activation of liver AMPK by 50 mg/kg metformin is contingent on both AXIN and LAMTOR1 implicating activation of the lysosomal pool. Likewise in intact mouse embryonic fibroblasts and in isolated vesicles from these fibroblasts incubated with metformin, the activation of AMPK was linked to a complex comprising the vacuolar H^+^-ATPase, Ragulator, AXIN and LKB1 and with reciprocal dissociation of mTOR/Raptor also implicating a predominant lysosomal pool. Recently Jia and colleagues [21,22] identified a role for cytoplasmic lectins termed galectins, which are recruited to the lysosome during stress resulting in mTOR inhibition (galectin-8) and AMPK activation by TAK1 (galectin-9) [21]. In the macrophage THP-1 cell line metformin induced the recruitment of galectin-9 and galectin-8, to a vesicular location [22]. Whether this effect is linked to metformin accumulation in lysosomes [44], inhibition of cathepsins [45] or a more pleiotropic effect of metformin on lysosomes remains to be determined.

### 2.5. AMPK Activation by A-769662, 991 and AMP Analogues Does Not Mimic Metformin on Gluconeogenesis

Small molecule activators of AMPK (A-769662 and 991) that bind to an allosteric site at the interface between the α-unit and the carbohydrate binding domain on the β-unit [46,47] or that are precursors of AMP analogues (C13 for C2 [48]) are potential tools to test the role of AMPK activation because they promote both allosteric activation of AMPK and phosphorylation of its downstream targets, and do not interact with FBP1 unlike AICAR that was used in prior studies. In hepatocytes they promote comparable phosphorylation of the AMPK substrate acetyl-CoA carboxylase (Ser^79^) as metformin but have converse effects on the partitioning of DHA between gluconeogenesis and glycolysis, on glycolysis from glucose, and on G6P levels during gluconeogenic flux [49,50]. Although non-specific effects of the compounds cannot be firmly excluded, the opposite effects from metformin on both gluconeogenic flux and cell G6P despite substantial AMPK activation, argues against a role for AMPK activation in the acute inhibition of gluconeogenic flux [49,50]. This does not exclude a role for AMPK on gluconeogenesis through effects on gene expression (Section 4).

## 3. Adaptive Changes in Hepatic Intermediates of Glycolysis and Gluconeogenesis

### 3.1. Effects of Nutritional State, Glucagon, Fatty Acids and Exercise on F1,6-P_2_

Control of intracellular metabolite homeostasis has been less extensively studied and is less well understood than blood glucose homeostasis [51]. The liver presents unique challenges for intracellular homeostasis because it receives the products of carbohydrate and protein digestion directly from the gut via the portal vein and is thus exposed to higher and more variable concentrations of sugars (glucose and fructose) and amino acids than peripheral tissues. A key role of the liver is the clearance of these substrates for energy interconversion and storage. Efficient control of substrate clearance is uniquely served by liver-specific enzyme isoforms.

Allosteric control of glycolysis in the liver is by feed-forward activation by substrate supply. This contrasts with feed-back inhibition in other tissues which is controlled by metabolic demand. Feed-forward activation of hepatic glycolysis by high glucose after a meal involves: (1) glucose-induced activation of glucokinase by dissociation from its inhibitory protein GKRP in the nucleus and translocation to the cytoplasm; (2) elevation in hepatic G6P and F6P; (3) F6P-induced activation of the bi-functional enzyme PFKFB1, which generates and degrades the regulatory metabolite, F2,6P_2_; (4) allosteric activation of PFK1 by the raised F2,6P_2_; (5) elevation in F1,6P_2_ the PFK1 product which in turn activates pyruvate kinase. Figure 1A,B shows the relative concentrations of the intermediates of glycolysis-gluconeogenesis in rat liver in the fed and fasted state [52]. The first 3 intermediates G6P, F6P and F1,6P_2_ are approximately 2-fold higher in the fed state consistent with feed-forward activation of glucose disposal. The concentrations of the substrate and products of aldolase (F1,6P_2_, DHAP,Ga3P) are substantially lower than more distal metabolites.

In liver F1,6P_2_ is the product of PFK1 during glycolysis and of aldolase during gluconeogenesis. This intermediate shows adaptive changes that mostly parallel the activity of PFK1 (Figure 1C,D). For example, glucagon phosphorylates the bifunctional enzyme PFKFB1 that generates and degrades F2,6P_2_, causing a rapid decline in F2,6P_2_ (activator of PFK1 and inhibitor of FBP1) with a slower decline in F1,6P_2_ [54] and concomitant elevation in G6P and F6P (Figure 1C,D). Likewise the decline in F1,6P_2_ in response to fatty acids (Figure 1E, [55]) can be explained by high-affinity inhibition of PFK1 by acyl-CoA [56]. The converse changes in F1,6P_2_ and F6P/G6P during exercise (Figure 1F, [57]) also implicate regulation at PFK1/FBP1. Although F2,6P_2_ determines PFK1 regulation by glucose and glucagon [58] other allosteric effectors of PFK1 (including AMP, Pi and citrate) determine the stimulation of glycolysis by anoxia [59] and metformin [49,60] which occurs despite a decline in F2,6P_2._

### 3.2. AICAR Has a Unique Effect on F1,6P_2_

AICAR (5-amino-4-imidazole carboxamide riboside), which is phosphorylated to ZMP, an intermediate in purine biosynthesis, is a strong inhibitor of gluconeogenesis and has a unique effect on intermediary metabolites characterized by a large increase in F1,6P_2_ and DHAP (Figure 2A), when the cytoplasmic ZMP is in the high mM range, because ZMP inhibits FBP1 with a Ki of 370 μM by binding to the AMP inhibitor site [61]. The fold increase in F1,6P_2_ is far greater with DHA (dihydroxyacetone) compared with lactate and pyruvate (40-fold vs. 2-fold) as substrate, consistent with the higher gluconeogenic flux from DHA [62]. It is noteworthy that whilst AICAR almost totally inhibits FBP1 as shown by the 40-fold increase in F1,6P_2_ with DHA in hepatocytes, it is not a selective inhibitor of FBP1 and also inhibits glycolysis in hepatocytes and other cell types in part by inhibition of PFK1 [63,64]. In addition because of sequestration of phosphate in ZMP, AICAR also lowers ATP by depleting inorganic phosphate and thereby oxidative phosphorylation [65]. It is therefore unsuitable as a tool for either AMPK activation or for FBP1 inhibition.

### 3.3. Similarities in Cross-Over Plots for Biguanides and Respiratory Chain Inhibitors

The inhibition by guanidine derivatives of mitochondrial respiration with substrates of Complex 1 has been recognized for ~60 years [66,67]. Although the therapeutic relevance was questionable, detailed flux control analysis by Halestrap and colleagues showed that the respiratory chain has a very high flux control coefficient on gluconeogenic flux from lactate and pyruvate [68], meaning that inhibition of glucose production can occur at a very small fractional decrease in oxidative phosphorylation that may not manifest as a significant decline in cell ATP. Owen & Halestrap [69] determined the effects of different inhibitors of the mitochondrial respiratory chain on intermediates of glycolysis/gluconeogenesis in rat hepatocytes and showed raised PEP, 3PG, 2PG and lowered triose phosphates (DHAP,Ga3P), F1,6P_2_, F6P, G6P (Figure 2B, red data curve). This is a distinct crossover plot from that of AICAR, but is identical to that of phenformin [70] in rat liver perfused with three gluconeogenic precursors (lactate/pyruvate, DHA, xylitol). Owen & Halestrap proposed that the decrease in ATP/ADP inhibits the phosphoglycerate kinase/Gapdh equilibria [69]. Their later work with metformin on rat liver in vivo [6] showed broadly similar results except for triose phosphates and F1,6P_2_ which were not decreased (Figure 2B, blue data curve). Lowering of triose phosphates and F1,6P_2_ by respiratory chain inhibitors [69] and phenformin [70] was associated with a substantial (>50%) decline in the ATP/ADP ratio, whereas the lack of lowering of triose phosphates and F1,6P_2_, by metformin in vivo was not associated with a decline in the total (free+bound) ATP/ADP ratio [6]. This suggests that the “target site” for inhibition of gluconeogenesis is at PFK1/FBP1 for modestly compromised phosphorylation potential, with unchanged or raised F1,6P_2_ [6], but at PGK/Gapdh with a decline in F1,6P_2_ for more severe inhibition of the respiratory chain and phosphorylation potential [69,70]. Accordingly, there is no evidence for unequivocal lowering of F1,6P_2_ by a modest metformin dose [6].

**Figure 2 ijms-21-03240-f002:**
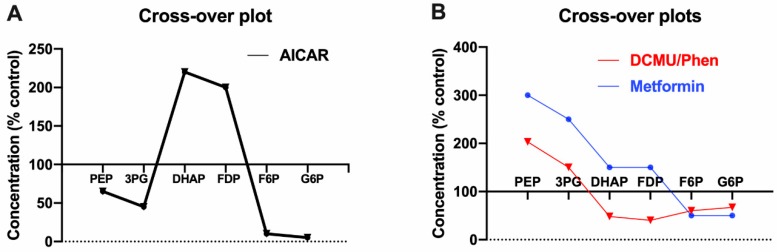
Crossover plots of metabolites of gluconeogenesis and glycolysis in liver or isolated hepatocytes. (**A**) Effects of AICAR (500 μM) in hepatocytes incubated with 10 mM lactate + 1 mM pyruvate [61]. (**B**) Effects of mitochondrial inhibitors (DCMU, dichlorophenyl dimethylurea) [69] or phenformin [70], in rat hepatocytes incubated with 10 mM lactate + 1 mM pyruvate (red) or effects of metformin on rat liver in vivo [6] (blue).

Other studies on the control of gluconeogenesis in the fetal/neonatal transition also showed crossover points at either aldolase or PFK1/FBP1 [71]. Crossover plots at PFK1/FBP1 make teleological sense insofar that PFK1 and FBP1 are highly regulated by allosteric effectors [58]. Support for FBP1 in the metformin mechanism was provided by Hunter and colleagues [72] who generated a mouse model for AMP-insensitive FBP1 and showed compromised efficacy of metformin (250 mg/kg) in this model. This data does not exclude composite effects of metformin on both FBP1 and PFK1. Hepatocyte studies targeting the PFK1/FBP1 locus by either depletion of F2,6P_2_ with a kinase-deficient variant of PFKFB1 or using aurintricarboxylic acid, a potent inhibitor of PFK1 (that mimics citrate inhibition) provide support for a role for PFK1 in the metformin mechanism [49,50]. Allosteric effectors of PFK1 and FBP1 that are known to be regulated by metformin in a direction consistent with a crossover point between F6P and F1,6P_2_ include: AMP (raised), Pi (raised), citrate (lowered), and G3P (lowered) [49,50]. There is limited metabolite data for the direct AMPK activators (A-769662, 991) that bind to the allosteric site. However, these compounds unlike metformin do not lower G6P and neither do they mimic the effects of metformin on the redox state [49,50].

### 3.4. Metformin Has a Biphasic Effect on the Mitochondrial NADH/NAD Redox State

Two independent groups reported inhibition of Complex 1 (NADH:ubiquinone oxidoreductase) by high millimolar metformin, in hepatocyte suspensions and mitochondria from metformin-treated rats [5,6]. This was supported by an increase in the 3-hydroxybutyrate/acetoacetate ratio, which indicates an increase in the mitochondrial NADH/NAD ratio through the 3-hydroxybutyrate dehydrogenase equilibrium [73], and confirmed in later studies [74,75]. Bridges and colleagues [7] used purified Complex 1 and assays for 3 sequential steps of the catalytic cycle (NADH oxidation, intramolecular electron transfer and ubiquinone reduction) and showed that metformin is a reversible, non-competitive inhibitor for the ubiquinone reduction reaction, with an IC_50_ in the millimolar range (~20 mM). Although this is ~300 fold higher than the [metformin] in the hepatic portal vein and ~100-fold higher than the concentration in the liver [76], metformin is estimated to accumulate in the matrix of energized mitochondria to 100-fold higher concentrations [7]. Accordingly the calculated matrix metformin at a portal vein concentration of 60 μM [76] is close to the IC_50_ [6,7]. Nonetheless, this issue remains contentious because subcellular fractionation studies do not support such high intramitochondrial concentrations [44,77]. However, cells pre-loaded with ^14^C-metformin, lose metformin very rapidly when transferred to metformin-free medium [49,60], and likewise mitochondria lose metformin during the isolation procedure [6]. Accordingly, the calculated mitochondrial content [6,7] is a more valid estimate than measurements after subcellular fractionation, and is not far removed from measurements of ^14^C-metformin in hepatocyte monolayers washed rapidly [49,60], if one assumes a fractional mitochondrial volume of 10% [78] and a 100-fold gradient [7].

Recent studies by Madiraju and colleagues [79,80], showed that a 3–5 fold lower dose of metformin (50 mg/kg) than was used by Owen & Halestrap [6] caused a more oxidized mitochondrial NADH/NAD redox state as determined from both the liver and plasma 3-hydroxybutyrate/acetoacetate ratio. This novel seminal finding was later corroborated by studies on the renal mitochondrial redox state which is more oxidized after 50 mg/kg metformin [81] but more reduced after 125 mg/kg metformin [82]. Our recent studies on mouse and rat hepatocytes showed that metformin has a biphasic effect on the mitochondrial redox state such that 100 μM metformin which corresponds to a cell load of 1–2 nmol/mg protein (therapeutic equivalent) causes a more oxidized whereas a 5-fold higher concentration cause a more reduced redox state [50]. This confirms that the more oxidized mitochondrial redox state in vivo after a low metformin dose [79,80] is a direct effect of metformin on the liver.

### 3.5. Candidate Mechanisms for the Oxidized Mitochondrial NADH/NAD State

Madiraju and colleagues proposed that the more oxidized mitochondrial NADH/NAD redox state at low metformin is explained by direct inhibition of mitochondrial glycerophosphate dehydrogenase (mGPD) [79,80], a component of the respiratory chain that spans the inner membrane and transfers reducing equivalents from G3P on the cytoplasmic side to ubiquinone in the respiratory chain, bypassing Complex 1 [83]. In liver, mGPD activity is low compared with other tissues [83]. Nonetheless changes in mGPD activity (by protein overexpression or irreversible inhibition) in hepatocytes associate with corresponding changes (more reduced or more oxidized, respectively) in the mitochondrial NADH/NAD redox state [50]. However to date, the inhibition of mGPD by therapeutic levels of metformin remains contentious because it has not been replicated in three other laboratories [50,84,85], the most recent of these used four independent mGPD assays [85]. Since mGPD is regulated allosterically by calcium [83], an indirect effect of metformin through allosteric or covalent mechanisms cannot be unequivocally excluded. Nonetheless in the absence of replication of metformin inhibition of mGPD activity, other mechanisms need to be considered.

An alternative mechanism is uncoupling of respiration. Cameron and colleagues [86] proposed compromised coupling of the redox and proton pumping domains of Complex 1 by metformin to explain a more oxidized NADH/NAD state in liver mitochondria. Batandier and colleagues compared metformin and rotenone and noted mild uncoupling by metformin (but not by rotenone) as determined from increased oxygen consumption with concomitant inhibition of ROS production linked to reverse electron transport [87]. The latter has been replicated in other studies with low metformin [88,89] and is attributed to an indirect pleiotropic effect of metformin resulting from a decrease in the driving force for proton pumping against the proton gradient [89].

### 3.6. Metformin Effects on the Cytoplasmic NAD/NADH State and Glycerol 3-P (G3P)

Metformin (at low or high doses) causes a more reduced cytoplasmic NAD/NADH redox state as determined from the lactate/pyruvate ratio [5,6,79,80,90]. The effect of high metformin (associated with a reduced mitochondrial state) has been attributed to inhibition of Complex 1 [5,6] while the effect of low metformin (associated with a more oxidized mitochondrial redox state) was attributed to inhibition of mGPD [79,80] which transfers reducing equivalents from the cytoplasm to mitochondrial ubiquinone pool (via the glycerophosphate shuttle) [83]. However in liver, the malate aspartate shuttle has a more prominent role in transfer of reducing equivalents than the glycerophosphate shuttle because of the low activity of mGPD [50,83,85,91]. The malate aspartate shuttle involves an electrogenic transmitochondrial exchange of aspartate for glutamate and H^+^ [92] which is very sensitive to mitochondrial membrane potential [93,94]. Accordingly, the more reduced cytoplasmic NAD/NADH by low metformin is consistent with mitochondrial depolarisation [89,95] although an increase in membrane potential has also been reported [77].

Liver cell G3P is either increased [79,80] or decreased by metformin [49,50,96]. Candidate effectors of changes in G3P are: the NAD/NADH_cyt_ redox state; the activity of mGPD and substrate supply (Figure 3). Cytoplasmic glycerophosphate dehydrogenase (cGPD), catalyses the reversible NAD/NADH coupled interconversion of G3P and DHAP, and accordingly a more reduced cytoplasmic NAD/NADH ratio predicts an increase in the G3P/DHAP ratio. The mitochondrial enzyme mGPD catalyses the irreversible conversion of G3P to DHAP coupled to transfer of electrons to the ubiquinone pool in the respiratory chain and accordingly G3P is decreased by overexpression of mGPD and also by mitochondrial uncoupling [49,50]. Substrate supply also increases G3P either moderately as occurs at high glucose or more substantially with fructose (8-fold) and xylitol (80-fold) [97,98,99]. Changes in G3P by metformin may therefore result from multiple possibly opposing mechanisms.

### 3.7. The Metformin Mechanism on Gluconeogenic Flux: Deductions from Cell Metabolites

Proposed mechanisms for the acute inhibition of gluconeogenic flux by metformin include: (i) activation of AMPK [3]; (ii) the decrease in the ATP/ADP ratio acting on the glyceraldehyde phosphate dehydrogenase and phosphoglycerate kinase equilibria [6,69]; (iii) an increase in the cytoplasmic NADH/NAD ratio acting on the lactate dehydrogenase equilibrium decreasing the cytoplasmic and mitochondrial pyruvate [6,79,80]; (iv) allosteric control at FBP1 and PFK1 which are regulated by multiple activators or inhibitors including AMP, Pi, citrate and G3P [49,50,58,72].

AMPK activation can be discounted for the rapid mechanism(s) because selective activators of AMPK do not mimic metformin on gluconeogenic flux [11,49,50]. The decrease in the ATP/ADP ratio would have a role in conditions of substantial ATP depletion as occurs with phenformin and mitochondrial inhibitors with a crossover point between DHAP and 3PG (Figure 2B) but not with a metformin dose causing negligible ATP depletion with a crossover point between F6P and F1,6P_2_ [6]. The increase in the cytoplasmic NADH/NAD predicts greater inhibition of gluconeogenesis from reduced substrates such as lactate and glycerol compared with pyruvate and dihydroxyacetone. Madiraju and colleagues proposed that inhibition of mGPD accounts for inhibition of gluconeogenesis through a more reduced cytoplasmic NAD/NADH [79,80] and showed that knock-down of mGPD mimics the metformin effect on the cytoplasmic redox state in Sprague Dawley rats. In addition metformin inhibited gluconeogenesis from reduced (lactate, glycerol) but not from oxidized (pyruvate, DHA) substrates in hepatocytes from Sprague Dawley rats. Two comments are warranted. First, most if not all other hepatocyte studies show higher basal rates of gluconeogenesis from DHA than from either lactate or glycerol, regardless of substrate concentration [62], contrary to the lower rates from DHA relative to lactate [79]. Second, inhibition by metformin of gluconeogenesis from lactate and glycerol was greater than 50%. Given that ATP depletion by metformin is greater with lactate and pyruvate than with glucose [23], and it is also greater with reduced than oxidized substrates [50], it is possible that ATP depletion accounts for the inhibition with reduced substrates [79]. Calza and colleagues [100] argued against the redox mechanism based on inhibition of gluconeogenesis from lactate with ethanol but not 30 μM metformin in the perfused rat liver [100].

To simultaneously test the redox hypothesis and the PFK1/FBP1 site in the metformin mechanism we used oxidized (DHA) and reduced (xylitol and glycerol) substrates and compared metformin with an inhibitor of the malate aspartate shuttle, which promotes a more reduced cytoplasmic NADH/NAD redox state compared with metformin [50]. We found modest inhibition of gluconeogenesis with DHA in the absence of ATP depletion but trends of ATP depletion with metformin in combination with the reduced substrates (glycerol and xylitol). The latter conditions are associated with raised G3P, which may explain the compromised ATP. This underscores the need for parallel analysis of cell ATP in all measurements of gluconeogenesis in isolated hepatocytes. Inhibition of the malate aspartate shuttle, which has a greater impact on the cytoplasmic NAD/NADH redox state, than low metformin, attenuated total xylitol metabolism [50], whereas metformin had a greater impact on the partitioning of substrate between gluconeogenesis and glycolysis. This supports a predominant role for metformin control at the PFK1/FBP1 site relative to a redox mechanism. Whilst these studies do not exclude an effect of metformin on gluconeogenesis from lactate, which is discussed in detail elsewhere [101], they underscore the role of the PFK1/FBP1 site in the metformin mechanism.

Hunter and colleagues [72] tested the hypothesis that metformin inhibits gluconeogenesis by an analogous mechanism to AICAR by inhibiting FBP1 at the AMP allosteric site. They showed that a mouse model with an AMP-insensitive FBP1, is resistant to the acute blood glucose lowering effect of AICAR and of a metformin dose (250 mg/kg) that causes a 2-fold increase in the AMP/ATP ratio. This is consistent with a crossover point with metformin between F1,6P_2_ and F6P as was shown previously [6]. Another mechanism linked to the raised AMP by metformin is inhibition of glucagon signalling through inhibition of adenylate cyclase [102]. Accordingly, three candidate mechanisms link raised AMP with inhibition of gluconeogenic flux through glucagon signalling, inhibition of FBP1 and activation of PFK1.

## 4. Control of Gene Expression of Gluconeogenic Enzymes by Metformin and AMPK Activation

### 4.1. Interactions between AMPK Signalling and PKA Signalling in the Metformin Mechanism

A role for changes in gene expression in the metformin inhibition of gluconeogenesis is suggested by studies in human type 2 diabetes showing lack of acute inhibition of hepatic glucose production within 4 h of intravenous administration of metformin [103], implicating chronic changes in gene expression to account for metformin efficacy during chronic therapy [2]. Gene expression profiling of livers of obese db/db mice 2 h after administration of metformin (50 or 400 mg/kg) identified repression of *G6pc* as one of 12 genes regulated by metformin [104]. *G6pc* encodes glucose 6-phosphatase, which catalyses the final reaction in hepatic glucose production. This enzyme also has a crucial role in regenerating inorganic phosphate in conditions of raised G6P [51]. Gene expression of *G6pc* like *Pck1* (encoding phosphoenolpyruvate carboxykinase) is induced several-fold (>10-fold) by cell permeable cAMP-analogues and this induction is attenuated by metformin [11,105,106]. The lower mRNA levels for both *G6pc* and *Pck1* in hepatocytes from a mouse model with a gain of function mutation in the AMPK-γ1 subunit (D316A) with a ~2.5-fold increase in AMPK activity supports a role for AMPK activation in the repression of both *G6pc* and *Pck1* in basal non-stimulated conditions [107]. This attenuated *G6pc* and *Pck1* expression manifests in hepatocytes isolated from the mice but not in vivo [107]. Isolated hepatocytes from genetic or drug/diet-treated models are a powerful approach to track the hepatocyte “status” independently of secondary hormonal and substrate changes that may have an overriding effect *in vivo*. Candidate transcription factors for the AMPK-dependent repression of *Pck1* and *G6pc* include CRTC2 (CREB regulated transcription factor 2, also known as TORC2) and CBP (CREB binding protein), where CREB is the cyclic AMP response element binding protein activated by glucagon signalling and the down-stream transcription factor target PGC1-alpha, which itself regulates the expression of *G6pc* and *Pck1* [4,11]. Increased cAMP degradation by activation of PDE4 is another candidate mechanism by which AMPK activation could counteract *G6pc* and *Pck1* induction by cAMP signalling [108]. However, despite extensive evidence for counter-regulation by AMPK activation of cAMP-mediated induction of *G6pc* and *Pck1* [4], studies on hepatocytes from AMPK-deficient mice showed virtually identical *G6pc* repression by metformin (≥0.5 mM) as control hepatocytes [11].

### 4.2. Metformin Counter-Regulation of Glucose Induction of G6pc

A distinguishing feature between transcriptional regulation of *Pck1* and *G6pc*, is that whereas both genes are activated by cAMP-signalling only *G6pc* is induced by high glucose by mechanism(s) that are at least in part mediated by ChREBP, the Carbohydrate response element binding protein [60,109,110,111]. High glucose concentration causes activation of ChREBP by translocation from the cytoplasm to the nucleus and recruitment to its target genes. The metabolite trigger is not glucose itself, because gene induction is abolished by hexokinase inhibitors and enhanced by inhibition of G6P hydrolysis with a chlorogenic derivative, which markedly raises G6P [110,111]. Although G6P is often regarded a candidate trigger of ChREBP activation based on bioinformatic considerations [112], the metabolite mechanism is more complex because depletion of F2,6P_2_ abolishes the glucose induction of ChREBP target genes [111]. ChREBP is regulated by covalent modification as well as by metabolite control and an inhibitory role for AMPK activation was proposed (reviewed in [113]). However, more recent work suggests an inhibitory effect of raised AMP, which may contribute to some of the effects that were previously attributed to AMPK activation [113]. In rat hepatocytes, metformin counteracts the induction by high glucose of *G6pc* and other ChREBP target genes including *Pklr* and also *Txnip* and it inhibits the recruitment of ChREBP to both the *G6pc* and *Pklr* gene promoters [60]. These effects of metformin are associated with lowering of G6P which also occurs in hepatocytes from AMPKα1, α2-KO mice [49,60]. Although the exact mechanism by which metformin represses *G6pc* through AMPK-independent mechanisms [11] remains to be elucidated, a role for metabolite control of ChREBP recruitment to the *G6pc* promoter [60] through either raised AMP [113] or lowered G6P and F2,6P_2_ [60] is currently the most plausible hypothesis.

## 5. Perspectives

There is widespread consensus that the activation of AMPK by metformin first reported 20 years ago occurs at therapeutic doses of the drug [3,4] and is involved in the inhibition of lipogenesis [3,10]. However what is less clear is the contribution of AMPK activation to the suppression of hepatic glucose production. When asking the latter question, it is important to distinguish between the chronic effects of metformin on gluconeogenesis that are mediated by repression of the *G6pc* and *Pck1* genes resulting in lower protein levels as distinct from the rapid effects of metformin on gluconeogenic flux that are independent of changes in *G6pc* and *Pck1* at the protein level.

For the rapid mechanism(s) on gluconeogenesis, the cumulative evidence is that activation of AMPK does not inhibit gluconeogenic flux. Small molecule activators of AMPK that target either the allosteric activator site or the AMP site and do not target FBP1 [48] have modest but opposite effects to metformin on gluconeogenic flux [11,49]. The rapid but modest inhibition of gluconeogenic flux by metformin is best explained by changes in allosteric effectors of FBP1 and PFK1 [49,72]. These allosteric effectors include raised AMP which is an inhibitor of FBP1 and activator of PFK1 and raised Pi and lowered citrate which activate and inhibit respectively, PFK1 [58]. Whether the more reduced cytoplasmic NAD/NADH redox state caused by metformin has an additional role in restricting the metabolism of lactate or glycerol remains as yet contentious [79,80,100,101].

AMPK activation counteracts cAMP-mediated induction of both *G6pc* and *Pck1* [105,106]. However, there is also evidence for repression of *G6pc* but not *Pck1* by metformin through AMPK-independent mechanism(s) [11]. The question whether the latter AMPK-independent repression of *G6pc* by metformin is quantitatively important relative to other AMPK-mediated mechanisms is relevant to the potential therapeutic value of selective AMPK activators for blood glucose control in type 2 diabetes.

Several points are noteworthy regarding the metformin repression of *G6pc*. First, the link between metformin and *G6pc* repression has been identified by non-biased approaches in animal models of obesity and hyperphagia predisposed to diabetes [104]. Second, *G6pc* catalyses the final reaction in hepatic glucose production but also has a major role in maintaining intrahepatic metabolite homeostasis, as supported by the marked induction in animal models of hyperphagia such as the ob/ob mouse or elevated G6P [51,114]. Third, *G6pc* and *Pck1* are oppositely regulated by high glucose [110,111], and also by gluconeogenic precursors such as dihydroxyacetone and glycerol, which like high glucose induce *G6pc* but repress *Pck1* [111,115]. Because *G6pc* and *Pck1* are regulated co-ordinately by cAMP-linked mechanisms and likewise by AMPK activation [105,106], but oppositely by nutrient excess, irrespective of whether the substrate is glucose or gluconeogenic precursors [110,115], measurement of both genes may help discriminate between AMPK-dependent and metabolite linked mechanisms which are independent of AMPK [49]. Type 2 diabetes is commonly associated with nutrient excess which predicts compromised intracellular homeostasis. Recent studies have reported lack of efficacy of metformin in healthy non-diabetic people as well as in newly diagnosed well-controlled type 2 diabetes [116,117]. Although several factors may interact with the efficacy of metformin on endogenous glucose production, further work on the metformin mechanism should focus on cell and animal models with compromised intracellular metabolite homeostasis that more closely simulate the dysregulation in poorly controlled type 2 diabetes, to determine whether AMPK-independent repression of *G6pc* by metformin predominates in the compromised metabolic state.

## Figures and Tables

**Figure 1 ijms-21-03240-f001:**
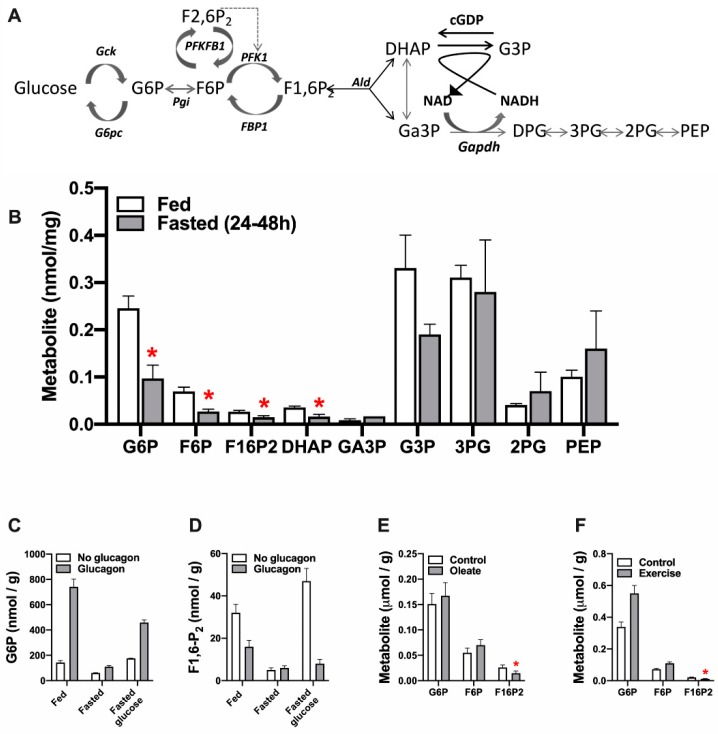
Liver concentrations of intermediates of glycolysis and gluconeogenesis. (**A**) Metabolic intermediates of glycolysis and gluconeogenesis. (**B**) Concentrations of key metabolites in rat liver in the fed and fasted state; data from Bergmeyer, HU [52]). (**C**,**D**) Liver metabolites in fed and fasted rats treated with glucose (2 g/kg, 10 min) or glucagon (1 mg/kg, 2 min); data from [53,54]. (**E**) Effects of 1 mM oleate on metabolites in hepatocytes from fasted rats, data from [55]. (**F**) Rat liver metabolites in rested and exercised rats, data from [56]. * *p* < 0.05 fasted vs. fed

**Figure 3 ijms-21-03240-f003:**
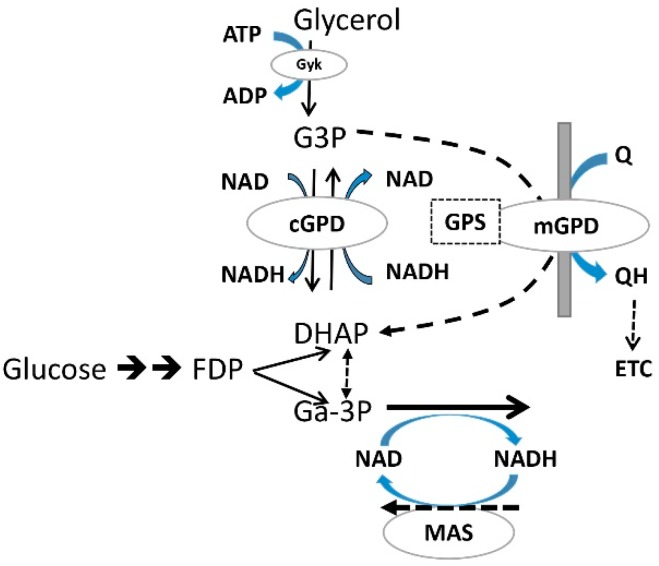
Metabolic pathways linked to glycerol 3-P formation in liver. G3P is generated from glycerol phosphorylation by glycerokinase or from DHAP, an intermediate of glycolysis and gluconeogenesis by cGPD, which catalyses the reversible NADH/NAD linked interconversion of DHAP and G3P; mGPD on the outer surface of the mitochondrial innermembrane catalyses irreversible oxidation of G3P coupled to the transfer of electrons to ubiquinone in the respiratory chain.

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
