# Peer review of "The Metformin Mechanism on Gluconeogenesis and AMPK Activation: The Metabolite Perspective"

_ijms, 2020, doi:10.3390/ijms21093240_

Round 1

Reviewer 1 Report

This review focuses on two recent publications by the senior author and coworkers (references 35 and 36 therein) with isolated hepatocytes and effects of metformin (M) on gluconeogenesis.

One central finding was that M, at between 0.1 and 0.2 mM, lowered G6P and repressed G6pc. Furthermore, these observations were replicated in hepatocytes from AMPK-KO mice and activators of AMPK were unable to mimic M. Evidence was presented that glycolysis was stimulated by M and although cytosolic NADH/NAD ratios indicated a more reduced state at 0.1 to 0.5 mM, the low concentrations of M (0.1 or 0.2mM) induced a more oxidized mitochondrial NADH/NAD+ ratio. The uncoupler 2,4-Dinitrophenol (DNP), the Complex I inhibitor rotenone and overexpression of mGPD mimicked the effects of M to some extent. As explanations, the authors suggest that M, by acting on mitochondria, is increasing the activity of PFK-1 and/or is inhibiting FBP-1 by changing allosteric modulators: AMP, phosphate, citrate and G3P.

Candidate explanations, according to the authors, are 1) inhibition of mGPDH 2) uncoupling of proton pumping by Complex I from electron transport or 3) mitochondrial depolarization at very low (i.e. <0.1 mM) concentrations of M.

Commentary

  • With respect to the mGPDH mechanism, the authors in reference 35, Figures 4a and 4b, and more detailed in the thesis of A. F. Alshawi, Section 4.2.4, Figure 4-8, reported no inhibition of mGPDH by M up to 5 mM. Furthermore, Pecinová et al., 2019 and Calza et al., 2018 presented evidence that M does not inhibit mGPDH. Overexpression of mGPDH had similar effects as M at low concentrations (Fig. 8 in reference 35). The final nail in the coffin of M inhibiting mGPDH is to be found here: MacDonald et al., 2020.
  • The conclusion of Cameron et al., 2018, that M is uncoupling, is based on a single observation in their Figure 8 with 1.9 and 10.9 mM concentrations and has not been confirmed by other researchers. It is interesting, however, that low concentrations of M (0.1 mM (Kane et al., 2010) or 0.001 mM (Robb et al., 2018) can inhibit peroxide production, when mitochondria are respiring on succinate.
  • As evidence for mitochondrial depolarization the authors apparently refer to Figure 7 of Dykens et al., 2008 which indeed shows a dose response curve for M and ΔΨ after 24 h incubation of hepatocytes. In the text, however, it is stated: “At 500 μM, metformin had no detectable effect on any of the indices, although it did dissipate ΔΨ and induce ROS production at 1 mM, and depletion of reduced glutathione was apparent at 2 mM.” In the text, six concentrations for each biguanide are listed but Figure 7 shows 7 concentrations. The caption of Figure 6 reads: ”Compared to vehicle control (top row), 24 h exposure to metformin, even at 500 μM, has little effect on any of the parameters, although it does so at 1 mM and higher concentrations (data not shown).” Thus, one can either believe the text or the figure. Others observed clear increases of ΔΨ in liver, muscle and kidney cells with 75 and 1000 μM M, respectively, after 22h (Wang et al., 2019).

Throughout the review, the authors challenge the view (see e.g. Section 2.4) that AMPK is involved in acute effects of M but admit later, in Section 4.1. that in T2D, there is no evidence for rapid inhibition of gluconeogenesis after i.v. injections of M as was observed 53 years ago with healthy rats or guinea pigs (Meyer, Ipaktchi and Clauser, 1967). As M is only active via the oral route, EGP (i.e. mainly HGP) in healthy subjects (McCreight et al., 2020) or in patients with recent-onset diabetes (Gormsen et al., 2019) is even increased by M.

The experimental conditions chosen (up to 45 mM glucose) are such that the basal activation state of AMPK is very low and mTORC1 is almost certainly not inhibited.

In the concentration-time window of 0.1s to 0.2 mM and 2 to 3 h the authors found no increased AMPK phosphorylation. It is almost common knowledge that longer incubation times, e.g. 39 h as employed by Zhou et al., 2001 allow much lower concentrations of M (10 μM) to stimulate AMPK. Similarly, 75 μM after 22 h was highly active Wang et al., 2019.

Commentary

The authors mention other pathways of AMPK activation (Section 2.1), namely control by aldolase and F1,6BP (references 12,13,14 in the review) and the role of lysosomes. Stimulated by the recent discussions about hydroxychloroquine/chloroquine (licensed in India for T2D treatment) in the context of COVID-19 (Gautret et al., 2020), the following is added:

The Rediscovery of the Late Endosomes/Lysosome as Metfomin Targets 

 In 2003, an interesting discovery was made (Sweeney, Raymer and Lockwood, 2003) (Lockwood, 2010). Several metal (e.g. Fe2+, Zn2+) complexes of guanidines, among them the antimalarial cycloguanil and the antidiabetics phenformin and M, inhibited cathepsin B proteolysis at very low concentrations. In a biological assay, employing rat hearts, perfused as Langendorff preparations, tissue proteins were first labeled with [3H]‑leucine for a short time, followed by chase with cold leucine. After 3 hours, when rapid protein turnover, as measured by release of trichloroacetic acid-soluble tracer, was low, biguanides with or without Zn2+ were added. It should be noted here, that even 2 mM M perfusion of Langendorff hearts does not stimulate AMPK or its downstream target ACC (Saeedi et al., 2008). The lysosomal subfraction of tracer release was defined by the maximal effect of the prototypical lysosomotropic agent chloroquine. M (25 µM) inhibits lysosomal proteolysis after 75 min about 50%, but 1 µM Zn2+ did not inhibit at all. However, when Zn2+ (1 µM) and M (25 µM) were combined, lysosomal proteolysis was rapidly and completely blocked. The authors concluded that “metformin greatly synergizes the anti-lysosomal action of extracellular Zn2+”. Lysosomal proteolytic inhibition by M has not been replicated despite an enormous scientific effort to elucidate the biology of lysosomes, with two exceptions: In their work on the primary mechanism by which M extends C.elegans lifespan, Chen et al., 2017 provided evidence that M in the growth medium inhibited Cathepsin B activity in the lysosomes of the worm. More recently, M after 2 h incubation at 100 and 250 μM inhibited Cathepsin B activity and induced a mild lysosomal perturbation, which activated the lysosomal damage response via the Gal9 sensing pathway, resulting in TAK1 phosphorylation of AMPK (Jia et al., 2020).

In view of the fact that AMPK activation in T2D patients treated with M has been observed in skeletal muscle, adipose tissue and circulating monocytes (Bhansali et al., 2020), which are never exposed to the extreme concentrations as the enterocytes in the gut, chronic but mild lysosomal damage and TAK1 as upstream kinase instead of invoking Complex I may play a major role.

In the context of the findings reported in the review, future focus could perhaps be on lysosomes/ endoplasmic reticulum instead on mitochondria.

References

Bhansali, S. et al. (2020) ‘Metformin upregulates mitophagy in patients with T2DM: A randomized placebo-controlled study’, Journal of Cellular and Molecular Medicine, 24(5), pp. 2832–2846. doi: 10.1111/jcmm.14834.

Calza, G. et al. (2018) ‘Lactate-induced glucose output is unchanged by metformin at a therapeutic concentration - A mass spectrometry imaging study of the perfused rat liver’, Frontiers in Pharmacology. doi: 10.3389/fphar.2018.00141.

Cameron, A. R. et al. (2018) ‘Metformin selectively targets redox control of complex I energy transduction’, Redox Biology, 14(June 2017), pp. 187–197. doi: 10.1016/j.redox.2017.08.018.

Chen, J. et al. (2017) ‘Metformin extends C. elegans lifespan through lysosomal pathway’, eLife, 6. doi: 10.7554/eLife.31268.

Dykens, J. A. et al. (2008) ‘Biguanide-induced mitochondrial dysfunction yields increased lactate production and cytotoxicity of aerobically-poised HepG2 cells and human hepatocytes in vitro’, Toxicology and Applied Pharmacology. Elsevier Inc., 233(2), pp. 203–210. doi: 10.1016/j.taap.2008.08.013.

Gautret, P. et al. (2020) ‘Clinical and microbiological effect of a combination of hydroxychloroquine and azithromycin in 80 COVID-19 patients with at least a six-day follow up: A pilot observational study’, Travel Medicine and Infectious Disease. Elsevier, p. 101663. doi: 10.1016/J.TMAID.2020.101663.

Gormsen, L. C. et al. (2019) ‘Metformin increases endogenous glucose production in non-diabetic individuals and individuals with recent-onset type 2 diabetes’, Diabetologia. Diabetologia, 62(7), pp. 1251–1256. doi: 10.1007/s00125-019-4872-7.

Jia, J. et al. (2020) ‘AMPK, a Regulator of Metabolism and Autophagy, Is Activated by Lysosomal Damage via a Novel Galectin-Directed Ubiquitin Signal Transduction System’, Molecular Cell. Elsevier Inc., 77(5), pp. 951-969.e9. doi: 10.1016/j.molcel.2019.12.028.

Kane, D. A. et al. (2010) ‘Metformin selectively attenuates mitochondrial H2O2 emission without affecting respiratory capacity in skeletal muscle of obese rats’, Free Radical Biology and Medicine. Elsevier Inc., 49(6), pp. 1082–1087. doi: 10.1016/j.freeradbiomed.2010.06.022.

Lockwood, T. D. (2010) ‘The lysosome among targets of metformin: new anti-inflammatory uses for an old drug?’, Expert opinion on therapeutic targets, 14(5), pp. 467–478. doi: 10.1517/14728221003774135.

McCreight, L. J. et al. (2020) ‘Metformin increases fasting glucose clearance and endogenous glucose production in non-diabetic individuals’, Diabetologia. Diabetologia, 63(2), pp. 444–447. doi: 10.1007/s00125-019-05042-1.

Meyer, F., Ipaktchi, M. and Clauser, H. (1967) ‘Specific inhibition of gluconeogenesis by biguanides.’, Nature, 213(5072), pp. 203–4. doi: 10.1038/213203a0.

Pecinová, A. et al. (2019) ‘Mitochondrial targets of metformin—Are they physiologically relevant?’, BioFactors, (March), p. biof.1548. doi: 10.1002/biof.1548.

Robb, E. L. et al. (2018) ‘Control of mitochondrial superoxide production by reverse electron transport at complex I’, Journal of Biological Chemistry, 293(25), pp. 9869–9879. doi: 10.1074/jbc.RA118.003647.

Saeedi, R. et al. (2008) ‘Metabolic actions of metformin in the heart can occur by AMPK-independent mechanisms’, American Journal of Physiology-Heart and Circulatory Physiology, 294(6), pp. H2497–H2506. doi: 10.1152/ajpheart.00873.2007.

Sweeney, D., Raymer, M. L. and Lockwood, T. D. (2003) ‘Antidiabetic and antimalarial biguanide drugs are metal-interactive antiproteolytic agents’, Biochemical Pharmacology, 66(4), pp. 663–677. doi: 10.1016/S0006-2952(03)00338-1.

Wang, Y. et al. (2019) ‘Metformin Improves Mitochondrial Respiratory Activity through Activation of AMPK’, Cell Reports. ElsevierCompany., 29(6), pp. 1511-1523.e5. doi: 10.1016/j.celrep.2019.09.070.

Zhou, G. et al. (2001) ‘Role of AMP-activated protein kinase in mechanism of metformin action.’, The Journal of clinical investigation, 108(8), pp. 1167–74. doi: 10.1172/JCI13505.

Author Response

We thank Referee-1 for the considerable time and effort spent on our Manuscript and for all the points raised.    We have endeavoured to incorporate as many of these as possible.

Commentary

  1. The mGPDH mechanism: The paper by MacDonald (31/03/2020) was published online after completion  of our manuscript.  Nonetheless we agree that it is a very important paper and refer to it and the study by Pecinova in the revised version.
  2. The study by Calza et al. (2018) did not measure mGPD, but is referred to elsewhere in connection with the cytoplasmic redox state. It is not clear whether the reported metformin concentration (30 uM) refers to the intracellular concentration or perfusate.   In the gavage studies with a therapeutic dose of 50 mg/kg (Wilcock & Bailey, 1994), the portal vein concentration was 50-60 uM but the total liver concentration (at peak levels) would be 3-fold higher.  It appears therefore that the concentrations used were submaximal.   Nonetheless we refer to the study by Calza et al. 2018 in the revised version in the other section on the mechanism.
  3. With respect to Cameron (2018), Kane (2010) and Robb (2018). Cameron’s conclusion is based on a single figure presented as oxygen consumption and NADH traces.   This work should be viewed in the context of the challenge posed in the understanding of Complex 1 function.  The team of Judy Hirst did a superb exercise in testing the metformin mechanism on the first 3 reactions of Complex 1 using the purified multiprotein complex, but this assay does not lend itself to vectorial proton transport.  The merit of  Cameron’s study showing a more oxidized redox state by metformin but not by DG8 should therefore be viewed in context.  The experiment in Fig.8 was performed in Hassinen lab in Oulu, a scientist with an extensive record in this methodology (Hassinen, IE, 1986, Methods Enzymol, 1986).   Uncoupling at Complex 1 or downstream raises the question of the fate of the electrons.  Here the literature is complex and contentious.  A decrease in ROS was reported by most but not all studies.  De Haes P (PNAS, 2014) reported an increase in ROS production in C. elegans after 24h exposure.  Whether the converse response is linked to exposure time is unclear.   Although Robb et al (2018) titrated FCCP they did not titrate metformin.  This is the only study to report metformin efficacy at 1uM and has not been replicated.  Nonetheless even if there a reporting error in the concentration of metformin in this study, the effect on membrane potential was depolarisation. Although Cameron’s study has not been replicated, prospectively, it concurs with previous work from Leverve’s laboratory (Batandier C. 2006) on isolated mitochondria prepared from livers previously perfused with metformin, mild inhibition of Complex 1 in conjunction with uncoupling and inhibition of ROS by reverse electron transport.   We refer to the studies by Kane (2010), Robb (2018), Batandier (2006) in the revised version. [87-89].
  4. Membrane potential: Dykens (2008); Wang (2019). The study by Dykens is one of several that reported mitochondrial depolarisation by metformin.  The study by Robb et al. (2018) also reported depolarisation by metformin in isolated liver mitochondria.  The study by Wang is the only one reporting the opposite. This was not a composite cell population assay, but a single cell measurement.  In addition it was a chronic rather than acute study and it is not clear how membrane potential was normalized on a single cell assay.  However, this is referred to in the revised version. 
  5. Study by Wang/He (2019). A key issue with this study which was testing the role of AMPK in the metforminmechanism is that all the experiments that used the AMPK-KO cells lacked controls: 6 shows no effect of metformin in the AMPK-KO but lacks lox/lox controls; Fig. 7 was entirely in the presence of metformin without a metformin free control (according to the Legend to Fig. 7).  Fig. Suppl 5 compares a 5-wk study (Panel F without metformin) with a 9-week study (Panel K entirely with metformin).    Although the legends suggest that were no appropriate controls in Figs 6 and 7 and S5, the Results Section is written otherwise.  This paper is very difficult to assess.
  6. McCreight et al (2020) and Gormsen et al (2019). These are both very interesting studies and were on non-diabetic subjects (McCreight) and on well controlled T2D (Gormsen).  These references are added to the revised version to the final Section.  We have been unable to access a copy of the paper by Meyer F, Ipatachi M, Clauser H (1967) and cannot therefore refer to it.  There are of course several possible explanations for the lack of efficacy of metformin in these studies (on healthy subjects and type 2 diabetes with good control) compared with the meta-analysis in Reference-2.  Possibilities include; (i) that metformin efficacy relates to the severity of hyperglycaemia and compromised intracellular homeostasis as supported by studies on preclinical models; (ii) to elevated glucagon; (iii) to extrahepatic effects of metformin.  Clinical studies do not allow a distinction between the 3 possibilities since these studies did not compare good glycaemic control vs poor glycaemic control, therefore any conclusions remain speculative.  It is noteworthy that metformin is ineffective in MODY-2 (MODY-GCK) diabetes which is characterized by raised blood glucose but not by raised G6P.  This and other evidence suggests that the severity of compromised homeostasis may be a key factor in the respose to metformin.  However, in the absence of studies testing this question specifically, any conclusion remain speculative.  A statement on this issue is added to the final section in the revised version.
  7. The study that used 45mM glucose (Fig. 2) also used 25mM glucose (Fig. 8) and the results were similar in both data sets. As stated in the paper, 45mM glucose was used because the cell G6P levels is intermediate between 25mM glucose without and with S4048.  The latter inhibitor raises G6P and thereby simulates compromised homeostasis, which is relevant for T2D with poor glycaemic control.
  8. Gautret (2020). Publications on Covid-19 are interesting  but are beyond the scope of our paper on metformin.
  9. We refer to the studies linking metformin to cathepsins in the revised version and to the studies by Jia et al (2020). We had access to the primary figures of this MS but not the supplementary figures.    Without this data the only evidence for the metformin experiments is the “puncta” from immunostaining.   Nonetheless this is an interesting study (for the work preceding the metformin expt) which links lysosomal damage to AMPK activation and we refer to this work in the revised version.  The limitation is the evidence for  what is meant by lysosomal damage with metformin.
  10. Zhou (2001); Wang (2019): We have attempted to clarify in the revised version that the focus of our paper is not whether metformin causes activation of AMPK. We believe there is good consensus that:  (i) metformin activates AMPK;  (ii) AMPK activation inhibits lipogenesis.  We do not dispute this.  The question is whether activation of AMPK controls gluconeogenesis.   The data shows fairly unequivocally that:  (i) AMPK activators do not inhibit gluconeogenesis acutely and thereby do not mimic metformin.   (2)  Repression of G6pc by metformin involves both AMPK-dependent and AMPK independent mechanisms.   The study by Wang et al (2019) lacks controls for the experiments in Figures 6 and 7  and S5 which deal with AMPK dependence.  The conclusion that can be drawn from this study is that following subcellular fractionation all the 14C-metformin is present in the cytoplasmic fraction.  This concurs with the conclusions of Owen and Halestrap (2000) that metformin is rapidly lost during  isolation of mitochondria [6] and with the caution in the Discussion by Wilcock & Bailey (1991) and also with the rapid decline in liver metformin content in vivo (Wilcock & Bailey, 1994) and with the calculations of Bridges et al. (2014).

We made several changes to the Manuscript and we hope that it is much improved, thank you for the suggestions.

Reviewer 2 Report

This is a very interesting and very well written review on the mechanisms of action of the glucose lowering drug metformin, with particular emphasis on the role of metabolic intermediates in this effect. Written by a team that made multiple contributions on the regulation of carbohydrate metabolism, the review extensively covers the subject, and in a balanced way. It will be an excellent source of information for all those who are interested by the subject. My only suggestion is that a few concluding remarks might be helpful to the reader. I would leave this to the appreciation of the authors.

Author Response

Referee-2

We thank the referee for the very helpful and encouraging comments.

A final perspectives section is added to the revised version

Reviewer 3 Report

The authors present a comprehensive and balanced review that addresses the highly-debated question of metformin action, from the perspective of AMPK and altered glycolytic/gluconeogenic flux. My only major suggestion is to add a brief summary to recap published mechanisms, current perspectives and remaining questions.

Minor points:

line 74: this is slightly misleading – structural information is only known for γ1. Domain interactions for large γ2 and γ3 N-terminal extensions, missing in γ1, are unknown. Better to say γ subunits are stabilised in the AMPK complex through an interdomain β-sheet formed with β-CTD…

line 78: ADP binding also promotes T172 phosphorylation by CaMKK2 and LKB1. Instead of reviews cite original references for ADP/AMPK regulation: Ross et al 2015 Biochem J, Xiao et al 2010 Nature, Oakhill et al 2011 Science.

line 139: the context of this sentence is unclear – please expand. What is meant by ‘chemical inhibitors’?

line 181: convoluted sentence.

Fig 3 legend: maintain consistency with GPDH/GDH abbreviations.

Author Response

Referee-3

We thank the referee for the time and effort and for the very helpful comments.

A final perspectives section is added to the revised version.

Line 74, corrected as recommended, thank you.

Line 78, amended and the 3 references are referred to in the revised version, thank you.

Line 139, corrected.

Line 181, reworded.

Fig. 3 and corresponding legend are corrected.